# Water conservation benefits of urban heat mitigation

Pouya Vahmani [1] & Andrew D. Jones[1]

Many cities globally are seeking strategies to counter the consequences of both a hotter and drier climate. While urban heat mitigation strategies have been shown to have beneficial effects on health, energy consumption, and greenhouse gas emissions, their implications for water conservation have not been widely examined. Here we use a suite of satellite-supported regional climate simulations in California to show that broad implementation of cool roofs, a heat mitigation strategy, not only results in significant cooling, but can also meaningfully decrease outdoor water consumption by reducing evaporative and irrigation water demands. Irrigation water consumption across the major metropolitan areas is reduced by up to 9% and irrigation water savings per capita range from 1.8 to 15.4 gallons per day across 18 counties examined. Total water savings are found to be the highest in Los Angeles county, reaching about 83 million gallons per day. Cool roofs are a valuable solution for addressing the adaptation and mitigation challenges faced by multiple sectors in California.

---

[1] Lawrence Berkeley National Laboratory, One Cyclotron Road, Berkeley, CA 94720, USA. Correspondence and requests for materials should be addressed to P.V. (email: pvahmani@lbl.gov)

Urban areas are at the forefront of climate mitigation and adaptation efforts given their high concentration of people, industry, and infrastructure[1]. Warming trends and their potential consequences for energy demand and public health in urban areas are of high concern around the world[2–7]. Climate change is also expected to increase water stress in urban areas across the globe, adding to the pressures posed by the growing population and the energy and agriculture sectors[8, 9]. With its drought-prone, semi-arid Mediterranean climate, and densely populated urban areas, California is at the forefront of developing solutions that increase resiliency in the face of higher temperatures and increased water scarcity. During 2012–2014, California experienced a progressive and persistent drought[10] constituting the most severe drought conditions in the past century[11]. To cope with the extreme and unprecedented drought conditions, the first statewide municipal water use restriction executive order was issued[12], mandating a reduction of 25% in urban water consumption. The water supply and demand imbalances in this region are only expected to intensify in the coming decades, due to higher likelihood of drought conditions[13–15], increased spring snowmelt[16], and accelerated population growth. Posing another challenge for California, the decade spanning 2001–2010 was warmer than any decade of the twentieth century in the Southwest region of the United States[17]. There is high confidence that this warming trend will continue, bringing longer and hotter heat waves throughout the twenty-first century[17].

As a response to these water scarcity and warming trends, several studies have evaluated the effectiveness of different urban water conservation measures such as landscape conversion, indoor water efficiency measures, and tiered water pricing on water demand[18–21]. Numerous studies, on the other hand, investigated the effects of heat mitigation strategies on health, energy consumption, and greenhouse gas emissions[22–26]. The effectiveness of cool roofs, in particular, has been broadly investigated as a promising heat mitigation measure[22] that shows potential to meaningfully decrease outdoor and indoor temperatures, reduce cooling loads, and offset $CO_2$ emission via negative global radiative forcing. There are compelling reasons to expect cross-sectoral impacts between water conservation and

heat mitigation strategies. A few studies, for instance, have recently investigated the effect of water conservation measures on heat mitigation[27, 30]. However, the implications of heat mitigation efforts for water conservation have not been widely examined.

In this study, we assess the benefits of widespread deployment of cool roofs, a heat mitigation measure, from a water conservation point of view by focusing on evaporative and irrigation water demands. To accomplish this aim, we employ a customized and validated version of the Weather Research and Forecasting Model (WRF), coupled with an urban canopy model (UCM). We incorporate high-resolution and real-time satellite-based information to improve the model representation of land surface physical characteristics including albedo and green vegetation fraction (GVF). We also incorporate and validate a realistic urban irrigation module to capture the fluctuations of outdoor water use and its' interactions with weather conditions. We conduct two series of high-resolution simulations representing control and cool-roof scenarios. Control simulations use satellite-derived surface albedos while cool-roof simulations represent widespread deployment of cool roofs by increasing all building roof albedos to those commercially achievable in the current cool roofs industry. Our results show that broad implementation of cool roofs, not only leads to significant cooling of air temperature, but also meaningfully reduces outdoor water use by decreasing evaporative and irrigation water demands.

## Results

**Impact of cool roofs on irrigation water demand.** We present our results for the warm, dry months of June–October for 15 years (2001–2015) over the most densely populated regions of Northern and Southern California (hereafter referred to as NorCal and SoCal, respectively), including the 18 counties that comprise the San Francisco, Sacramento, Los Angeles, and San Diego metropolitan areas (Fig. 1). Our analysis shows that implementing cool roofs results in 4–9% irrigation water savings across the 18 counties examined in California (Fig. 2). Los Angeles county shows, by far, highest water saving percentage (9.1%), followed by Orange (7.9%) and San Bernardino (7.3%)

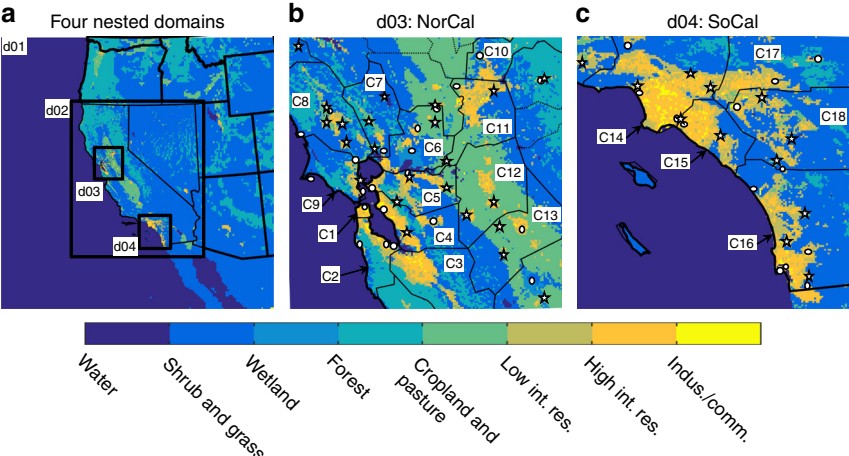

**Fig. 1** Land cover map and geographical representation of study domains. **a** Nested configuration of four WRF-UCM domains with 13.5, 4.5, 1.5, and 1.5 km spatial resolutions for d01, d02, d03, and d04, respectively. **b** Domain d03. **c** Domain d04. The solid black lines in **b**, **c** illustrate the boundaries of the 18 urban counties that are captured in domains d03 and d04: San Francisco (C1), San Mateo (C2), Santa Clara (C3), Alameda (C4), Contra Costa (C5), Solano (C6), Napa (C7), Sonoma (C8), Marin (C9), Placer (C10), Sacramento (C11), San Joaquin (C12), Stanislaus (C13), Los Angeles (C14), Orange (C15), San Diego (C16), San Bernardino (C17), and Riverside (C18). The location of NCDC and CIMIS stations, used in validation process, are indicated with circles and stars, respectively. Note that urban areas are represented through three land cover types: low intensity residential (Low int. res.), high intensity residential (High int. res.), and industrial/commercial (Indus./comm.).

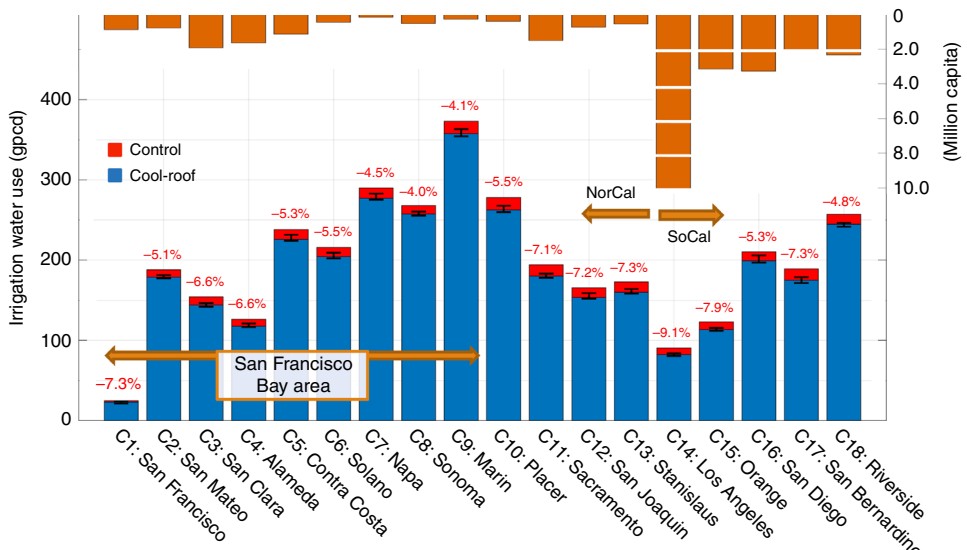

**Fig. 2** Simulated irrigation water consumption for control and cool-roof simulations for each county. Values represent averages in gallons per capita per day (gpcd) over urban surfaces for June–October of 2001–2015. The error bars illustrate inter-annual fluctuations of irrigation water consumption reductions induced by cool roofs. County populations are represented in orange bars. Note that daily rates of irrigation water use are calculated by dividing accumulated irrigation water over June–October of each year by 365, assuming irrigation happens only during these warm, dry months of the year

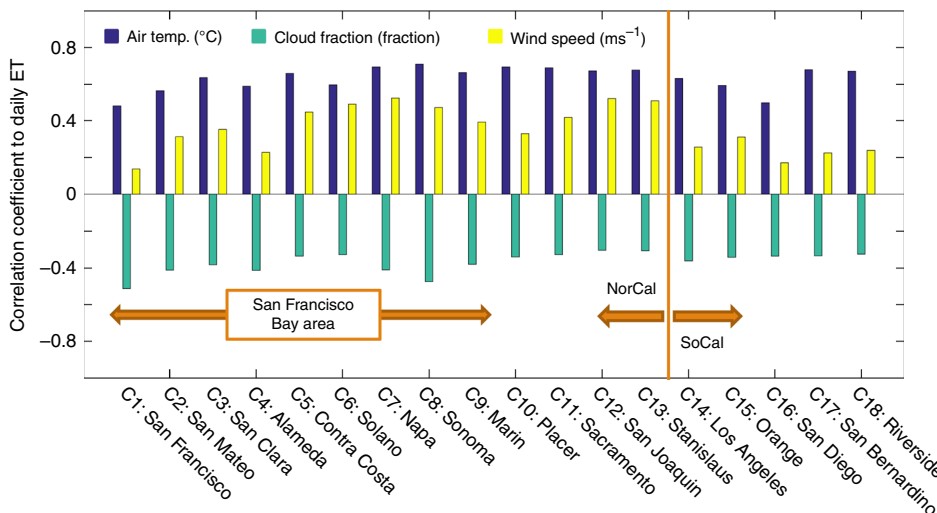

**Fig. 3** Variance of the daily evapotranspiration explained by different drivers. Correlation coefficients between simulated evapotranspiration (ET) versus air temperature, cloud fraction, and wind speed for conrol simulations. Values are calculated over urban surfaces of each county for June–October of 2001–2015

counties in SoCal and Stanislaus (7.3%), San Francisco (7.3%), San Joaquin (7.2%), and Sacramento (7.1%) in NorCal. We further show that Marin county in NorCal and Riverside county in SoCal have highest irrigation water demand per capita of 373 and 257 gallons per day, respectively, due to a high fraction of low-intensity residential areas and associated vegetation cover (urban fractions of 0.24 and 0.26, respectively). Highly developed counties of San Francisco (urban fraction of 0.62) and Los Angeles (urban fraction of 0.50), on the other hand, show lowest urban irrigation water use per capita of 24.7 and 90.6 gallons per day, respectively. Irrigation water savings per capita, induced by cool roofs, range from 1.8 to 15.4 gallons per day. Total water savings in Los Angeles are the highest (83 million gallons per day), due to the vast extent of this county. In fact, total cool-roof-induced water savings in the three southern counties of Los

Angeles, San Diego, and Orange (~150 million gallons per day) are nearly equivalent to the total water savings in other 15 studied counties in California combined.

**Drivers of evaporative water demand.** Evapotranspiration (ET), or evaporative water demand, is the main driver of simulated urban irrigation water consumption, explaining 91 and 92% of irrigation water variations across urban areas in NorCal and SoCal, respectively (Supplementary Fig. 1). Therefore, to better understand irrigation water demand and its response to cool roofs, we next focus on identifying the drivers of spatial and temporal variabilities of ET. Our results show that day-to-day variations of ET are dominantly correlated to air temperature fluctuations, which explain 23–50% of the daily ET variation

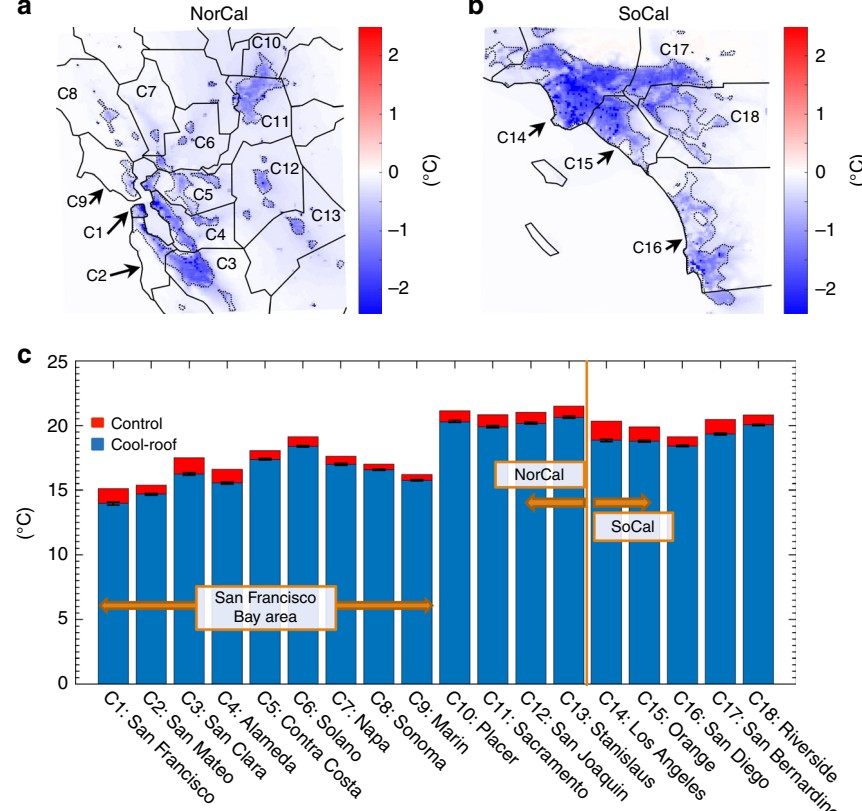

**Fig. 4** Simulated cool-roof-induced changes in 2-m air temperature. **a**, **b** Cool-roof-induced air temperature changes for NorCal and SoCal, respectively. **c** air temperature for control and cool-roof simulations for each county. Values represent averages over June–October of 2001–2015. The error bars in **c** illustrate inter-annual fluctuations of air temperature changes induced by cool roofs. The solid black lines in **a**, **b** illustrate the boundaries of the 18 urban counties that are captured in the model domains. The boundaries of urban surfaces are illustrated by dotted black lines. Note that only changes that are statistically distinguishable from zero at 95% confidence interval are included

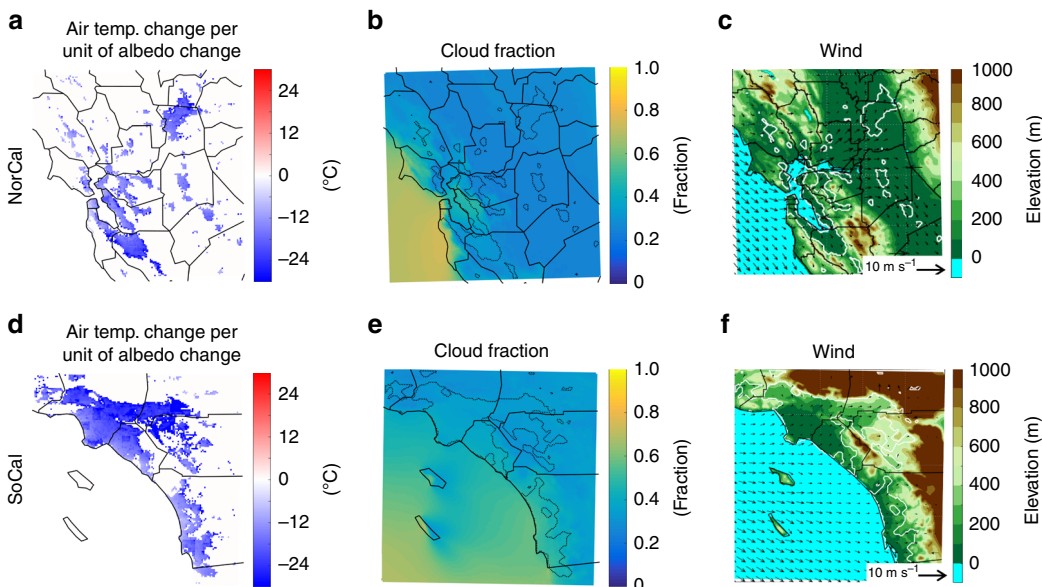

**Fig. 5** Spatial distribution of cool-roof-induced changes in air temperature per unit of albedo change and its drivers. **a**, **d** Simulated cool-roof-induced changes in 2-m air temperature per unit of albedo change. **b**, **e** Simulated cloud fraction and **c**, **f** 10-m wind vectors over elevation map for control simulations. Values represent averages over June–October of 2001–2015 for NorCal (**a**, **b**, **c**) and SoCal (**d**, **e**, **f**). The solid black lines illustrate the boundaries of the 18 urban counties that are captured in the model domains. The boundaries of urban surfaces are illustrated by dotted black lines in **a**, **b**, **d**, **e** and by white lines in **c**, **f**. Note that only changes that are statistically distinguishable from zero at 95% confidence interval are included in **a**, **d**

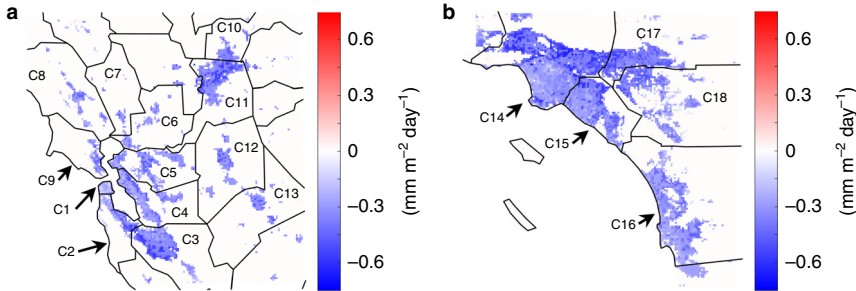

**Fig. 6** Spatial distribution of cool-roof-induced changes in evaporative water demand. Simulated cool-roof-induced changes in evaporative water demand for NorCal (**a**) and SoCal (**b**). Values represent evapotranspiration averages over June–October of 2001–2015. The solid black lines illustrate the boundaries of the 18 urban counties that are captured in the model domains. Note that only changes that are over urban areas and statistically distinguishable from zero at 95% confidence interval are included

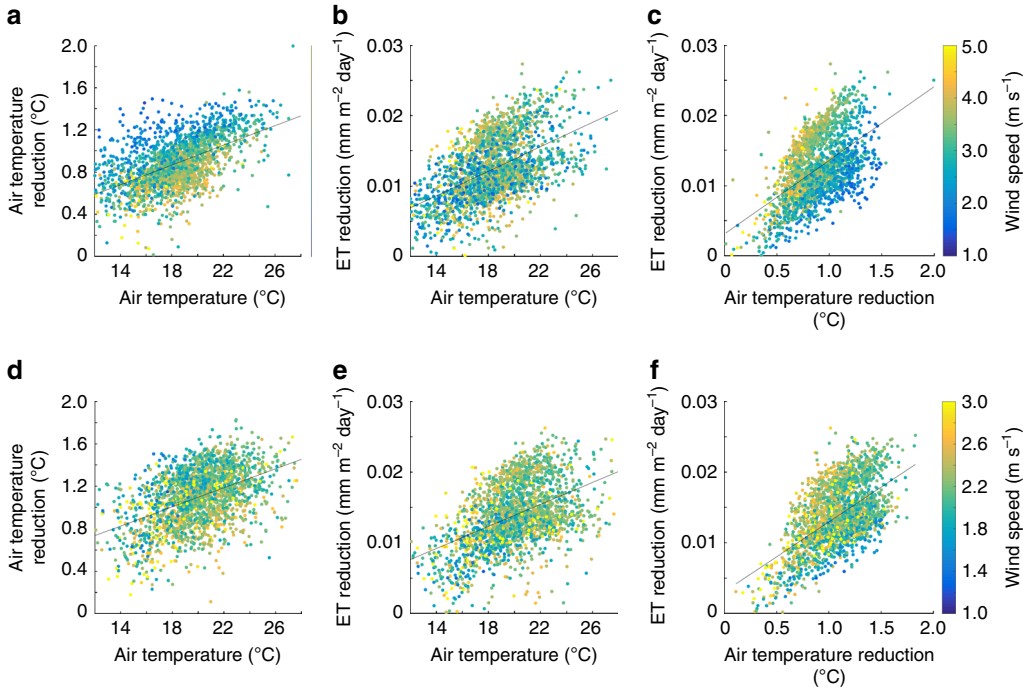

**Fig. 7** Correspondence between absolute air temperature and cool-roof-induced reduction in air temperature and evapotranspiration. Circles represent simulated daily means of air temperature reduction versus air temperature (**a**, **d**), evapotranspiration (ET) reduction versus air temperature (**b**, **e**), and ET reduction versus air temperature reduction (**c**, **f**) for June–October of 2001–2015, averaged over urban surfaces in NorCal (**a**, **b**, **c**) and SoCal (**d**, **e**, **f**). The circle colors along with the color bars illustrate the daily wind speeds. The black lines show the best linear fit (in a least-squares sense) to the data

across 18 counties in NorCal and SoCal regions (Fig. 3 and Supplementary Figs. 2–4). Cloud fraction and wind speed also play considerable roles, explaining 4–26% and 2–28% of day-to-day ET fluctuations. Cloud fraction shows negative correlation with ET as high cloud cover limits incoming solar radiation, surface energy budget, and therefore latent heat flux and ET rates. This effect is most significant over the fog-prone San Francisco Bay counties. This analysis also shows that increased wind speed results in increased ET. This relationship is the strongest over inland counties (e.g., Napa, San Joaquin, and Stanislaus), where wind transports hot and dry air into urban regions (Supplementary Fig. 5), providing energy for ET and, at the same time, increasing evaporative water demand by reducing vapor pressure (i.e., partial pressure of water vapor) in the atmosphere.

**Cooling effects of cool roofs**. The strong relationship between daily evaporative water demand and air temperature, reported

above, illustrates that cooling effects of heat mitigation measures, such as cool roofs, can in turn lead to reduced ET and evaporative water demand and therefore reduced irrigation water use. Our analysis shows that the widespread deployment of cool roofs results in a mean urban cooling of 1.0 and 1.2 °C over NorCal and SoCal, respectively (Fig. 4). The strongest cooling signal is seen over Los Angeles county (1.5 °C), followed by Santa Clara (1.2 °C), San Francisco (1.1 °C), San Bernardino (1.1 °C), Orange (1.1 °C), and Alameda (1.1 °C). Among major counties (population of 1 million or more), San Diego and Contra Costa show the smallest cooling signal (~0.7 °C). The cooling effects of cool roofs reported here are in agreement with the previous studies in the Los Angeles area reporting cooling signals of 1–2.5 °C[25, 29]. Albedo change patterns (Supplementary Fig. 6) are the dominant determinant of the cooling responses to the cool roofs ($R^2 = 0.77$ and 0.74 for NorCal and SoCal, respectively). For instance, significant albedo increases and therefore cooling signals are experienced over San Francisco county and downtown Los

Angeles, due to high urban fraction and industrial/commercial building cover over these counties. Assessing air temperature cooling per unit of albedo increase, we show that the cloud cover and extent of urban regions along with wind speed and direction also affect the spatial patterns of the cooling response to the cool roofs (Fig. 5). Over the northern portions of the San Francisco Bay area, where the prevailing delta breeze brings low-level cloud and fog inland through a gap in the coastal mountains (e.g., Alameda and Contra Costa counties), as well as foggy western side of San Francisco, the cooling effects of cool roofs weakens. Higher cloud cover over San Diego, relative to other SoCal counties, also results in the smaller air temperature cooling per albedo change. Over inland areas with less cloud cover, wind patterns play an important role by spreading the cooling signal from cool roofs over the downwind non-urban areas, leading to a weaker local cooling effects. This effect is more significant over narrow urban regions such as Solano, San Joaquin, Stanislaus, and parts of Riverside (see Fig. 4a, b). On the other hand, less cloudy regions with broad urban extents where localized air circulation is bound by topography (e.g., Los Angeles, San Bernardino, and Santa Clara) experience an accumulated cooling effect of cool roofs.

**Response of evaporative water demand to cool roofs**. The increased surface albedo, induced by cool roofs, results in increased reflected solar radiation and therefore decreased surface energy budget, decreased turbulent fluxes, and finally lower ET or evaporative water demand rates. Our results show that implementing cool roofs over NorCal and SoCal leads to average urban evaporative water demand reductions of 15 and 18%, equivalent of 0.36 and 0.38 mm day$^{-1}$, respectively (Fig. 6). Across the 18 counties examined, the most heavily populated counties of San Francisco, Santa Clara, Alameda, Los Angeles, and Orange (with total population of 17.6 million) show the highest evaporative demand reduction percentages of 23, 16, 17, 19, and 18%, respectively. These localized differences are in part due to the distribution of urban versus vegetated areas and cooling signal variation from cool roofs (Fig. 4 and Supplementary Fig. 7) across the region. Our analysis (Supplementary Fig. 8) shows a complicated relationship between ET reduction versus urban fraction and air temperature reduction. This is because urban fractions are associated with two opposite effects on ET reductions. Higher urban fractions lead to higher albedo change and stronger cooling signals (Supplementary Fig. 6), which increases ET reductions per unit of irrigated area. However, higher urban fraction also means less vegetated cover to respond to cooler temperatures. These counteracting forces result in a weak correlation between ET reduction versus air temperature reduction and urban fraction, when all urban types are considered. However, over low-intensity residential areas, where there is a significant vegetated cover to reflect the temperature changes in ET rates, a strong positive correlation exists between air temperature reduction and ET reductions. The greatest absolute reductions in ET are found in medium density areas with an urban fraction of ~0.4. This suggests that although cool-roof-induced air temperature reductions are most significant over highly urbanized regions, medium density regions benefit more in terms of reduced evaporative water demand. It is noteworthy that we assume the entirety of pervious surfaces in urban areas is irrigated. This assumption might affect the role of urban fraction, as discussed above, in determining baseline and reduced ET rates.

Our analysis further demonstrates that the cooling effects and evaporative water savings, induced by cool roofs, are more significant during the hotter days of the year (Fig. 7a, b, d, e) when ET levels are highest in the baseline. Air temperature and ET changes are considerably correlated with baseline fluctuations in daily air temperature with coefficient of determinations of 31% (21%) and 27% (21%), respectively, for NorCal (SoCal). Furthermore, wind speed plays an important role in partitioning the effect of cool roofs toward ET reductions versus air temperature reductions, particularly in NorCal. For a given baseline daily temperature, windier days are associated with smaller temperature reductions and larger ET reductions and the slope of the relationship between ET change and temperature change is much steeper (more ET change per unit temperature change) on windier days (Fig. 7c, f). By increasing albedo, cool roofs reduce the total amount of energy available to drive both latent and sensible heat fluxes. This result indicates that windier conditions shift this relationship toward a greater impact on the latent heat fluxes associated with ET. This shows that cool roofs are most effective, in reducing ET, over hot windy days, when highest evaporative water demands are experienced.

**Discussion**

We illustrate that cool roofs, a heat mitigation strategy, can be effective in both cooling the climate and reducing outdoor water use, but direct reductions in irrigation water use can have a contrary effect on heat mitigation efforts. Landscape conversion in urban settings has recently been shown to be one of the most cost-effective ways of conserving water[18]. Yet, an emerging literature on the topic demonstrates that such measures can have the unintended outcome of enhancing the urban heat island effect[27, 30], which is in turn associated with indirect costs such as increased cooling energy consumption[6], degraded human thermal comfort[28], heat-related mortality[28], and deterioration of air quality[31]. Moreover, the climatic consequences of such strategies can push back heat mitigation efforts designed to counter the effects of projected warming climate. A recent study[27] showed that completely stopping irrigation over Los Angeles metropolitan area results in daytime warming of up to 1.9 °C during the summer, largely due to shifts in surface energy partitioning toward higher sensible and lower latent heat flux. We tested this hypothesis over the San Francisco Bay Area over the warmer months of the year, June–October for 2012–2014. Our results show that reducing irrigation water increases air temperature. In the most extreme case, a complete cessation of irrigation leads to a mean daytime warming of 1.0 °C, averaged over the entire San Francisco metropolitan area (Supplementary Fig. 9). These results show that the warming signal from strategies that focus only on outdoor water use reductions can meaningfully offset the cooling effects of a major heat mitigation strategy, such as citywide cool-roof deployment.

The evaporative and irrigation water demand reductions reported are consistent over the 15 years examined including both drought and non-drought years, and hold to varying degrees across the four major metropolitan regions of California, which span 18 counties with significant variation in terms of geography, urban morphology, and microclimate including fog patterns, local air circulation patterns, and mean baseline temperatures. Significant urban and climatic variation across 18 counties considered offers insight into how these results might translate to other urban regions and into future climates. We see the strongest effects of cool roofs on both temperature and irrigation water use in regions with reduced cloud cover and widespread urban development. These areas also experience warmer peak temperatures and more significant outdoor water use in the baseline. We further show that cool roofs have the highest potential to increase surface albedo and therefore reduce air temperature over highly developed regions, but that medium density regions (urban fraction of ~0.4) with more vegetation benefit more in terms of

reduced water demand. We also find that the effect of cool roofs is greatest during the hottest days of the year, indicating that they could play an even greater role in reducing outdoor water use in a hotter future climate.

Despite the diversity of counties considered in this study, additional work is needed to explore the potential water conservation benefits of urban heat mitigation in alternative geographic and climate contexts. In particular, the lack of irrigation-season precipitation in Mediterranean climates like California's means that we did not need to consider the effect of cool roofs on precipitation, despite the potential interactions between precipitation and large-scale cool-roof deployment reported in previous studies[26, 32]. In addition, we note that more work is needed to explore the role of irrigation technology, behavior, and policy in mediating the relationship between reduced evaporative demand and reduced irrigation water applied. Although we calibrate and validate our irrigation scheme, the absolute and reduced irrigation water consumptions reported here could be influenced by a larger set of driving factors than the more direct effects of heat mitigation on ET. For instance, urban irrigation depends on behavior of land managers, vegetation type, irrigation technology, socio-economics, water pricing, and mandatory/voluntary restrictions, which are extremely difficult to predict and subject to significant changes.

Urban areas face growing multisector coordination challenges as they must simultaneously adapt to and attempt to mitigate the diverse effects of climate change. Our results point to the value of considering such efforts within a broader multisector climate adaptation and mitigation context. This is the first study to shed light on the potential of heat mitigation strategies to meaningfully contribute to urban water conservation efforts at a regional scale. In doing so, we identify a previously unrecognized strategy for significantly reducing regional water consumption, and add to the growing list of heat mitigation co-benefits, which already includes beneficial effects on health, energy consumption, and greenhouse gas emissions[22]. On the other hand, our results necessitate a cautionary note that direct irrigation water reductions have the potential to undermine heat mitigation efforts.

## Methods

**WRF-UCM modeling system**. We configure a satellite-supported version of WRF (version 3.6.1)[33, 34] over four nested domains with spatial resolution of 13.5, 4.5, 1.5, and 1.5 km. The two inner domains include San Francisco and Sacramento metropolitan areas in NorCal and Los Angeles and San Diego metropolitan areas in SoCal (Fig. 1). WRF is a state-of-the-art, fully compressible, non-hydrostatic, mesoscale numerical weather prediction model. To account for urban physical processes, WRF is coupled with a UCM[35, 36]. The single layer UCM, used in this study, treats the surface energy balance for urban areas, taking into account the three-dimensional nature of built surfaces, shadowing, reflections, and trapping of radiation as well as wind profile within an urban canyon[37].

We define the WRF-UCM initial and boundary conditions based on the North American Regional Reanalysis (NARR) data set[38]. The physics parameterizations utilized in the current study include the rapid radiative transfer model[39] for longwave radiation, the Dudhia scheme[40] for shortwave radiation, University of Washington (TKE) Boundary Layer Scheme[41] for the planetary boundary layer, the Morrison double-moment scheme[42] for microphysics, Grell-Freitas scheme[43] for cumulus parameterization (for domains 1 and 2), and the Eta Similarity scheme[44] for the model surface layer.

Urban fraction, which partitions each urban grid cell into pervious (undeveloped/vegetated) and impervious (developed) fractions, along with urban type defines the roof area and thus potential cool-roof area of each urban grid cell. On the other hand, urban fraction determines the non-urban or irrigated fraction of urban grid cells. For an accurate representation of cool roofs and urban irrigation, we replace the default and coarse USGS-based land cover and urban-type maps with the high-resolution (30 m) National Land Cover Data (NLCD)[45]. We further use the high-resolution (30 m) NLCD impervious surface data[46] instead of default urban-type-dependent tabulated urban fractions. We also incorporate the National Urban Database and Access Portal Tool (NUDAPT)[47] data set to define domain-specific urban morphology over the study domain.

It is noteworthy that the default WRF uses an unvarying sea-surface temperature. This is a problem particularly for long simulations, where it is

unrealistic to use the initial sea-surface temperature throughout the simulation. However, the model provides an alternative method where it takes time-varying sea-surface temperature as an input. Due to importance of sea-surface temperature fluctuations in the meteorology of the coastal cities in our study domain and significant sea-surface temperate change expected during 5-month-long simulations in this study, we incorporate the daily real-time sea-surface temperature (RTG_SST) product from the National Centers for Environmental Prediction/Marine Modeling and Analysis Branch (NCEP/MMAB).

**WRF-UCM simulations**. To investigate the cool roofs impacts on outdoor water consumption, we designed two series of high-resolution simulations representing control and cool-roof scenarios. For each scenario, 15 simulations are conducted from 20 May, 0700UTC (12:00 am local standard time) to 31 October, 0700UTC (12:00 am local standard time) for 2001 through 2015, over four two-way nested domains and 30 vertical layers. Including a spin-up of 10 days, our analysis is carried out over the warmer months of the year, June–October.

For the cool-roof scenario, the widespread deployment of cool roofs is represented by using increased roof albedos over all the buildings within our study domains, which replace the MODIS-based albedo values used in the control scenario. For a realistic simulation of cool roofs, we specify two types of cool roofs that are commercially available for industrial/commercial and residential buildings based on the EPA Energy Star roof product list (http://downloads.energystar.gov/bi/qplist/roofs_prod_list.pdf?8ddd-02cf), we prescribe the roof albedos of commercial/industrial and residential buildings to 0.85 and 0.60, respectively. The commercial/industrial cool-roof albedo of 0.85 is selected to reflect the highest roof albedo achievable by coating (i.e., 0.88 for SOLARFLECT coating). Although our intention is to assess the maximum potential impacts associated with widespread deployment of cool roofs, we choose a less aggressive value of 0.6 for residential cool-roof albedo to account for esthetic preference of the residents. It is noteworthy that the cool-roof albedo values used in this study are reported by the EPA Energy Star to be achievable after 3 years of wear and tear.

In the control and cool-roof scenarios, irrigation water is applied to the pervious portion of all the urban grid cells. Urban irrigation is accounted for by using a previously developed and validated irrigation scheme[30, 48], based on a soil moisture-deficit function. In each irrigation event, moisture content of the top soil layer (with depth of 10 cm) is set to a reference volumetric soil moisture threshold, below which transpiration begins to stress. Irrigation events are set to occur at nighttime (midnight) to avoid unrealistically heavy evaporation rates due to direct sun exposure. We design the irrigation scheme to mimic the common urban irrigation behavior in the sense that it happens at a predefined interval of three times per week. At the same time, we assume an efficient irrigation system that monitors soil moisture to avoid significant overirrigation or surface runoff, which might occur in practice. Note that daily rates of irrigation water use are calculated by dividing accumulated irrigation water over June–October of each year by 365, assuming irrigation happens only during these warm, dry months of the year.

To assess the statistical significance of the simulated changes, relative to model internal variability and natural variabilities of the climate system, we apply the two-sided Student's $t$ test to the model results. Analyzing the daily variations in each grid cell for the multiyear ensemble members, we only include the results that are statistically significant with a 95% confidence level in the presented maps.

**Satellite-based characterization of land surface in WRF-UCM**. Although the WRF-UCM is widely used to simulate urban climate, deficiencies exist with prescribed land surface physical characteristics. The mean climatological information or tabulated values, implemented in WRF, do not represent real-time state of the land surface. Furthermore, the information on LAI, GVF, and albedo either, do not apply to the urban grid cells or are misinterpreted for urban surfaces. For instance, the default model uses predefined and unvarying albedo and GVF values for vegetated portion of urban grid cells. The default WRF also incorrectly uses the input pixel-level LAI as pervious level LAI over urban surfaces. A recent study[49] explores these shortcomings in more details.

In the current study, we use real-time remotely sensed data to describe albedo and GVF for pervious and impervious urban surfaces as well as non-urban surfaces. We also apply the correct pervious level LAI over the pervious portion of urban grid cells. A comparison of the default climatological albedo, GVF, and LAI with the MODIS-based improved values is presented in the Supplementary Figs. 10 and 11.

MODIS observations are obtained from the U.S. Geological Survey (USGS) National Center for Earth Resource Observations and Science (EROS) website at http://earthexplorer.usgs.gov. We acquire domain-specific monthly maps of albedo and GVF based on MODIS reflectance (MCD43A3) and vegetation indices (MOD13A3), respectively. For LAI, as the MODIS-based LAI products (e.g., fraction of photosynthetically active radiation (MCD15A3) product) include significant amount of missing data over urban areas, we used tabulated LAI which relies on the improved MODIS-based GVF for inter-annual and monthly variabilities. The remotely sensed albedo and GVF maps with spatial resolution of 500 m and 1 km, respectively, are re-gridded to the WRF-UCM coordinate system and resolution.

One difficulty with using satellite-based albedo products over the urban areas is associated with the fact that these products provide one pixel-level value for each

grid cell. However, there are two portions, pervious and imperious, within each grid cells with different albedos. We modified the WRF-UCM framework to assign the remotely sensed pixel-level albedo to both pervious and impervious portions of urban grid cells. Thus, instead of using one albedo value for all the urban surfaces, we use measured domain-specific and spatially resolved values for both urban and non-urban areas. It should be noted that the same MODIS-based albedo value is used for all the urban surfaces within an urban grid cell (i.e., road, wall, and roof).

One important benefit of using real-time satellite-based characterization of the land surface in the current study is capturing the inter-annual fluctuations of these WRF-UCM inputs, which would be missed by the default climatological information on the land surface. Our improved forcing inputs show significant inter-annual fluctuations of GVF (Supplementary Fig. 12b, e, h), which in turn partially defines annual variabilities of albedo (Supplementary Fig. 12a, d, g) and LAI (Supplementary Fig. 12c, f, i). For instance, our results illustrate that the real-time MODIS GVF data captures the significant greening of the land in 2011 after a very wet year in 2010[50]. The effects of the recent California drought are also captured in the remotely sensed data as browning of the land from 2012–2015, induced by dry years of 2011–2015[50] and decreased irrigation encouraged by the state municipal water use restriction policies and recommendations[12]. Our results show that these fluctuations are most significant over natural and agricultural areas (Supplementary Fig. 12a–f). However, considerable fluctuations are observed over urban areas as well. It is also illustrated that increases (decreases) in GVF lead to increases (decreases) in LAI and decreases (increases) in albedo, respectively.

**WRF-UCM sensitivity to the initial conditions**. The sensitivity of our modeling framework to the initial conditions is assessed using three ensemble members. We repeated the control simulation for 2001 with three different initial start times: 20 May, 10 June, and 1 July. Our results show that three ensemble members converge after a few days (after 1 July) (Supplementary Fig. 13). This shows that the climate of the studied regions is mainly controlled by the large-scale atmospheric circulation patterns rather than local internal variability, affected by initial conditions. Moreover, the effects of initial soil moisture conditions are limited over irrigated urban regions, which are the focus of this study, as irrigation forces convergence of soil moisture in all ensemble members.

**Model validation**. We compare the predicted air temperature, ET, and added irrigation water with the ground-based observations to assess the model performance, after implementing satellite-based information as well as high-resolution NLCD-based land cover type and urban fraction. For air temperature, we use hourly ground measurements of 2-m air temperature from 35 stations from National Climatic Data Center (NCDC) network over our study domains for June–October of 2001–2015 (Fig. 1). And, for ET, we use the hourly estimates of reference ET ($ET_0$) for June–October of 2001–2015 from 34 California Irrigation Management Information System (CIMIS) stations distributed over our study domains (Fig. 1). CIMIS stations measure hourly meteorological data over well-watered actively growing closely clipped grass fields and use these measurements to estimate hourly reference ET (http://www.cimis.water.ca.gov/Resources.aspx). The implemented irrigation scheme has been previously validated over Los Angeles metropolitan area[30, 48]. We further use water consumption records from six parks (irrigation-only consumers) provided by the Contra Costa Water District to validate the simulated added irrigation water over our NorCal domain. Areal images of the parks locations are provided in the Supplementary Fig. 14. To account for the inevitable inconsistency between point measurements at the ground and the model results that are mean values over 1.5 × 1.5 km grid cells, we use averages of the measurements from all the stations as well as parks and compare them with averages over the corresponding grid cells.

Our validation analysis shows that the WRF-UCM predicts the daily mean (max) air temperatures reasonably well, with RMSDs of 1.4 °C (0.8 °C) and 1.2 °C (0.7 °C) for NorCal and SoCal, respectively (Supplementary Fig. 15). Comparing pervious level ET from WRF-UCM to the CIMIS-based reference ET observations (Supplementary Fig. 16), we show that the model reproduces the observed reference ETs also with a reasonable accuracy (RSMD of 0.67 and 0.74 mm day$^{-1}$). It is noteworthy that although the CIMIS stations are continuously irrigated, the urban landscapes, which include both grass and trees, have higher leaf area index and lower stomata resistance, relative to grass fields in CIMIS stations, making up for the lower soil moisture levels. Finally, we compare the model-added irrigation water accumulated over the five simulation months (June–October) with the outdoor water readings from irrigation-only consumers over the same months for each year (Supplementary Fig. 17). Considering the uncertainties associated with urban irrigation, our results show that the model predicts the amount of irrigation water over 15 years of simulations reasonably well (bias of −12%). The inter-annual variabilities in the simulated irrigation water reflect the fluctuations in the air temperature and therefore evaporative water demand. The more significant inter-annual fluctuations in the observed irrigation water consumption, on the other hand, are affected by many factors such as water conservation policies, water availability, and park management decisions in each year which are not reflected in the model. It is noteworthy that the model assumes a smart and efficient irrigation system that monitors the soil moisture to avoid excessive irrigation. The conservative nature of simulated irrigation system matches the irrigation behavior

in field with a better accuracy toward the latter years of 2001–2015 when the water conservations efforts intensify (Supplementary Fig. 17).

**Code availability**. The WRF model source code, documentation, and other resources can be found at http://www2.mmm.ucar.edu/wrf/users/.

**Data availability**. All the satellite data used in this study can be downloaded from the USGS National Center for EROS website at http://earthexplorer.usgs.gov. Other relevant data in this study are available from the authors per request.

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

## Acknowledgements

This work was supported by the Laboratory Directed Research and Development Program of Lawrence Berkeley National Laboratory under U.S. Department of Energy Contract No. DE-AC02-05CH11231. This research used resources of the National Energy Research Scientific Computing Center, a DOE Office of Science User Facility supported by the Office of Science of the U.S. Department of Energy under Contract No. DE-AC02-05CH11231. We also like to thank Contra Costa Water District for providing irrigation water consumption data used in the validation processes.

## Author contributions

P.V. led the writing, design of the paper, and analysis with guidance and mentorship of A.D.J. Both authors contributed to the text.

## Additional information

**Competing interests:** The authors declare no competing financial interests.

