## [Peer Review File · Nature Communications]

Reviewers' comments:

Reviewer #1 (Remarks to the Author):

Review for – Nature Communications Manuscript Number: NCOMMS-17-00243-T
Title: Water conservation benefits of urban heat mitigation

The authors conduct high-res simulations with WRF coupled to an urban canopy model for three separate summers (2012-2014) that correspond to the recent Bay Area drought. They use a simple irrigation scheme and conduct additional simulations with cool roofs (for the same period of time) to quantify their damping effect on near-surface temperatures, for the Bay Area, and the further impact of irrigation cessation. They calculate the impact of cool roof usage on ET demand and argue that their focus on sectors beyond temperature (for example, the water sector) makes this contribution particularly novel.

In the view of this referee, I don't find this manuscript to be of sufficiently broad relevance to justify publication in Nature Communications in its current state. For example, the applicability of cool roofs, only, omits additional strategies; the focus on one region (rather than more regions) prevents constructive dialogue about how communities may make use of these results, since the conclusions are Bay Area specific. In addition, the sample size (only 3 drought summers, for 1 region), limited ensemble size (1 member for each scenario), etc., raises robustness concerns. Therefore, the conclusions drawn are relevant for 1 area, for 1 strategy, based on simulations conducted for 1 meteorological regime. Finally, considerable research has already demonstrated implications on ET of a range of such strategies, but the narrow literature review performed selectively omitted many of them.

In my view, I would have liked to see a geographically based comparison of multiple regions at high-resolution, of multiple strategies that performs a space-based evaluation of what works best, and where. With additional simulations (which should include non-drought years as well, with the likely outcome that these differing hydro meteorological regimes would lead to different magnitude impacts on ET) that would include a range of geographies/cities, an extension beyond simply cool roofs, and a dedicated statistical assessment to convince the broad readership of the conclusions drawn, I could see this manuscript becoming suitable for Nature Communications.

Specific Comments

Lines 49...: "and heat mitigation strategies are assessed via their climatic impacts [8,9,10]. "

In making this statement, the authors neglect important recent work that has addressed such heat mitigation strategies that extend to other sectors. For example, Georgescu et al. (2014) examined the efficacy of several such strategies by extending the heretofore climate discussion to the energy sector, and additionally compared such effects to emissions of LLGHGs. Such attribution of the existing literature is necessary to ensure a balanced perspective is offered.

Georgescu, M., Morefield, P. E., Bierwagen, B. G., & Weaver, C. P. (2014). Urban adaptation can roll back warming of emerging megapolitan regions. *Proceedings of the National Academy of Sciences*, 111(8), 2909-2914.

Additional research has demonstrated the synergistic interactions of urban-induced warming and heat waves, and given the Introductory message the authors try to convey, several recent manuscripts come to mind as important to include as part of the discussion, including Li and Bou-Zeid (2013):

Li, D., & Bou-Zeid, E. (2013). Synergistic interactions between urban heat islands and heat waves: the impact in cities is larger than the sum of its parts*. *Journal of Applied Meteorology and Climatology*, 52(9), 2051-2064

Line 65: "and unsustainable groundwater extractions [18,19]."

The Sustainable Groundwater Management Act is aimed explicitly to address the unsustainable

groundwater extractions the authors assume will continue into the future. No matter your perceptions about AB1739, etc., this is a clear example of past extraction does not equal future extraction. The authors are encouraged to provide better-directed discussion that supports this important research.

Line 106: "... Fig. 1b and 1c ..."

Why do the authors present Figure 1b before Figure 1a; or Figure 2 before Figure 1a ... etc. The authors should pay attention to how they organized their figures such that it represents the actual flow of the manuscript.

Figure Captions: Several figure captions have misspellings (e.g., "temperture" in Figure 5).

Reviewer #2 (Remarks to the Author):

This paper studied an extremely important topic using well designed and thoroughly validated numerical experiments, which were built on the improvements made by the leading author into WRF-UCM. The idea is novel and is relevant to policy makers and the general public, and the results are interesting and understandable (even to the general public). The methodology is sound and the paper is well written. I like the paper very much and I think this is probably the first paper I ever reviewed for which I don't have any comments. Good job. I suggest publish as it is.

Reviewer #3 (Remarks to the Author):

This is a very interesting scientific work that merits to be published subject to the following improvements:

- a) Ageing effects have to be discussed and considered. How aging affects the results?
- b) An error and a sensitivity analysis have to be included.
- c) The results of the simulation have to be compared against similar simulations carried out for the same area, (Taha et al, Akbari, Menon, etc).
- d) The basic physics mechanism on the relation between high albedo and water conservation has to be better explained.

REVIEWER #1:

The authors conduct high-res simulations with WRF coupled to an urban canopy model for three separate summers (2012-2014) that correspond to the recent Bay Area drought. They use a simple irrigation scheme and conduct additional simulations with cool roofs (for the same period of time) to quantify their damping effect on near-surface temperatures, for the Bay Area, and the further impact of irrigation cessation. They calculate the impact of cool roof usage on ET demand and argue that their focus on sectors beyond temperature (for example, the water sector) makes this contribution particularly novel.

In the view of this referee, I don't find this manuscript to be of sufficiently broad relevance to justify publication in Nature Communications in its current state. For example, the applicability of cool roofs, only, omits additional strategies; the focus on one region (rather than more regions) prevents constructive dialogue about how communities may make use of these results, since the conclusions are Bay Area specific. In addition, the sample size (only 3 drought summers, for 1 region), limited ensemble size (1 member for each scenario), etc., raises robustness concerns. Therefore, the conclusions drawn are relevant for 1 area, for 1 strategy, based on simulations conducted for 1 meteorological regime. Finally, considerable research has already demonstrated implications on ET of a range of such strategies, but the narrow literature review performed selectively omitted many of them.

In my view, I would have liked to see a geographically based comparison of multiple regions at high-resolution, of multiple strategies that performs a space-based evaluation of what works best, and where. With additional simulations (which should include non-drought years as well, with the likely outcome that these differing hydro meteorological regimes would lead to different magnitude impacts on ET) that would include a range of geographies/cities, an extension beyond simply cool roofs, and a dedicated statistical assessment to convince the broad readership of the conclusions drawn, I could see this manuscript becoming suitable for Nature Communications.

We thank the reviewer for the constructive comments. We took every comment very seriously and used them to improve our study and paper. To address the robustness concerns, in particular, we extended our study domain to more regions, including all the major metropolitan areas in California rather than only San Francisco Bay area. We further conducted our experiments over 15 years of 2001-2015, rather than only drought years of 2012-2014. Doing so we have obtained 15 years worth of remotely sensed data to improve the representation of green vegetation fraction and surface albedo over urban areas. We further acquired irrigation water consumption, reference evapotranspiration, and air temperature data for 15 years for a comprehensive validation of our improved model, showing an accurate and reliable model performance. It is noteworthy that we did all these extra experiments while conserving the high spatial resolution of our modeling framework. We believe expanding the study domain and time in addition to our extensive efforts to represent the reality of the urban surfaces, vegetated fraction, and urban irrigation using real-time satellite information, irrigation water measurements, and high-resolution land cover maps have made are results not only more

robust and reliable but more interesting as we now include the variability of water saving potential of cool roofs over 18 counties and 15 years.

Although the main conclusions of the original manuscript hold true and we were able to show the robustness of our results, to reflect the new experiments, we have made substantial changes to our manuscript. We submit the revised manuscript with tracked changes and address each and every concern of the reviewer individually below.

“... For example, the applicability of cool roofs, only, omits additional strategies;...”

“For example, Georgescu et al. (2014) examined the efficacy of several such strategies by extending the heretofore climate discussion to the energy sector, and additionally compared such effects to emissions of LLGHGs.”

We understand the reviewer’s concern here. However, our purpose with this study is to use cool roofs as a representative heat mitigation strategy to shed light on an overlooked benefit of these strategies. Moreover, there are two types of heat mitigation strategies that are extensively studied and shown to be effective in urban areas: 1) technologies aiming to increase the albedo of cities and 2) the use of vegetative – green roofs [Santamouris 2014]. The study cited by the reviewer [Georgescu et. al. 2014] is also limited to these two types of heat mitigation strategies: cool roofs and green roofs. However, for the purpose of our study, which is water conservation benefits of heat mitigation, green/vegetated roofs are not a good fit as they actually increase water consumption.

Santamouris M. Cooling the cities – a review of reflective and green roof mitigation technologies to fight heat island and improve comfort in urban environments. *Sol Energy* 2014;103:682–703.

Georgescu, M., Morefield, P. E., Bierwagen, B. G., & Weaver, C. P. (2014). Urban adaptation can roll back warming of emerging megapolitan regions. *Proceedings of the National Academy of Sciences*, 111(8), 2909-2914.

“... the focus on one region (rather than more regions) prevents constructive dialogue about how communities may make use of these results, since the conclusions are Bay Area specific.”

We agree with the reviewer’s comment and extended our study domain from one metropolitan area to 4 major metropolitan areas: San Francisco, Sacramento, Los Angeles, and San Diego. The new study domain covers 18 counties that include coastal and inland areas in the northern and southern California (please see Fig. 1). By covering 4 major metropolitan regions and 18 counties over 15 years, we believe we have demonstrated a sufficiently robust result that would be of wide interest to both the scientific community and public sector practitioners involved in urban planning and water management.

Original:

Figure S1. (a) Land cover map and geographical representation of the three nested WRF-UCM domains with 13.5, 4.5, and 1.5 km spatial resolutions for d01, d02, and d03, respectively. (b) Land cover map of the inner most domain d03. The black lines in (b) illustrate the boundaries of the 9 counties in the San Francisco Bay Area. The location of NCDC and CIMIS stations are indicated with circles and stars respectively in (b).

Revised:

Figure 1. Land cover map and geographical representation of four nested WRF-UCM domains with 13.5, 4.5, 1.5, and 1.5 km spatial resolutions for d01, d02, d03, and d04, respectively. The black lines in (b) and (c) illustrate the boundaries of the 18 counties, urban areas of which are captured in domains d03 and d04: San Francisco (C1), San Mateo (C2), Santa Clara (C3),

Alameda (C4), Contra Costa (C5), Solano (C6), Napa (C7), Sonoma (C8), Marin (C9), Placer (C10), Sacramento (C11), San Joaquin (C12), Stanislaus (C13), Los Angeles (C14), Orange (C15), San Diego (C16), San Bernardino (C17), and Riverside (C18). The location of NCDC and CIMIS stations, used in validation process, are indicated with circles and stars, respectively.

In this revised manuscript, we showed the robustness of our results and provided a geographically based comparison of the ET and irrigation water savings induced by cool roofs for coastal and inland urban regions from northern to southern California, while preserving the high-resolution of our simulations. We also believe that the semi-arid Western US is an appropriate testbed region given its drought-prone climate and heavy reliance on water conveyance and storage systems to provide urban water. Nevertheless, there is limit to the extent of the urban areas we could cover, due to high computational cost of these high-resolution simulations over a period of 15 years (5 months per year). But, our goal here is also to start a conversation about, and encourage more studies to investigate, this overlooked co-benefit of heat mitigation strategies. We reflect this intention of ours in the manuscript:

L250: “Despite the diversity of counties considered in this study, additional work is needed to explore the potential water conservation benefits of urban heat mitigation in alternative geographic and climate contexts.”

“... In addition, the sample size (only 3 drought summers, for 1 region), limited ensemble size (1 member for each scenario), etc., raises robustness concerns.”

We agree with the reviewer’s comment. To address this issue, we extended our study period to cover 5 irrigation months (June-Oct.) of 15 years (2001-2015), including both drought and non-drought years, instead of drought years (2012-2014) only. We show the inter-annual variabilities of irrigation water savings and air temperature reductions, induced by cool roofs, with error bar on Figures 2 and 4. Although, the results do not seem to significantly change over drought versus non-drought years (see Fig. R1), we believe extending our study to 15 years has made our results more robust.

Figure R1. Simulated Cool-roof-induced changes in ET for 2001-2015. Values represent accumulative ET over June-Oct. of each year averaged over all urban surfaces in NorCal and SoCal.

To address the concern of the referee about ensemble size or the impact of initial conditions on the results we repeated the Control simulation for 2001 with three different initial start times: 20 May, 10 June, and 1 July. Our results show three ensemble members converge after a few days and nearly match after 1 July (Fig. S4). This shows that the climate of the studied regions, is mainly controlled by the large scale atmospheric circulation patterns rather than local internal variability, affected by initial conditions. Moreover, the effects of initial soil moisture conditions are limited over irrigated urban regions, which are the focus of this study, as irrigation forces convergence of soil moisture in all ensemble members. To address this important comment the Section 4 and Figure S4 are added to the supplementary information.

Supplementary L159:

“4. WRF-UCM sensitivity to the Initial Conditions

The sensitivity of our modeling framework to the initial conditions are assessed using three ensemble members. We repeated the Control simulation for 2001 with three different initial start times: 20 May, 10 June, and 1 July. Our results show three ensemble members converge after a few days (after 1 July) (Fig. S4). This shows that the climate of the studied regions, is

mainly controlled by the large scale atmospheric circulation patterns rather than local internal variability, affected by initial conditions. Moreover, the effects of initial soil moisture conditions are limited over irrigated urban regions, which are the focus of this study, as irrigation forces convergence of soil moisture in all ensemble members.”

Figure S4. Simulated hourly 2-m air temperature over July of 2001 for NorCal (a) and SoCal (b). The simulated averages over urban areas from 3 ensemble members with different initial start times are presented by black lines. Red fillings highlight the difference between simulated temperatures from these ensemble members.

“Finally, considerable research has already demonstrated implications on ET of a range of such strategies, but the narrow literature review performed selectively omitted many of them.”

We thank the reviewer for this comment. While ET is mentioned in the existing literature on urban heat islands (it is after all, a major component of the surface energy balance), this is the first study that focuses on the implications of urban heat mitigation strategies on irrigation water demand. The novel aspect of this study is using a validated irrigation scheme incorporated within a high-resolution and carefully calibrated regional climate modeling framework, to quantify and test the significance of cool-roof-induced irrigation water savings

associated with heat mitigation. As now stated in the manuscript, this result points to a previously unrecognized, yet promising strategy for cities to conserve water and also adds to growing list of urban heat mitigation co-benefits. That said, we thank the reviewer for suggesting additional literature to cite. We have included the studies suggested by the referee [Georgescu et. al., 2014; Li et al., 2013] and more [Georgescu et. al., 2012] in our revised manuscript:

L45: “While urban heat mitigation strategies have been shown to have beneficial effects on health, energy consumption, and greenhouse gas emissions [26,8,9,10,37], their implications for water conservation have not been widely examined.”

L85: “Numerous studies, on the other hand, investigated effects of heat mitigation strategies on health, energy consumption, and greenhouse gas emissions [26,8,9,10,37].”

L65: “Warming trends and their potential consequences for energy demand and public health in urban areas are of high concern around the world [21-25, 38].”

L254: “... the potential interactions between precipitation and large-scale cool roof deployment reported in previous studies [37;40].”

[37] Georgescu, M., Morefield, P. E., Bierwagen, B. G., & Weaver, C. P. (2014). Urban adaptation can roll back warming of emerging megapolitan regions. *Proceedings of the National Academy of Sciences*, 111(8), 2909-2914.

[38] Li, D., & Bou-Zeid, E. (2013). Synergistic interactions between urban heat islands and heat waves: the impact in cities is larger than the sum of its parts*. *Journal of Applied Meteorology and Climatology*, 52(9), 2051-2064

[40] Georgescu M, Mahalov A, Moustou M (2012) Seasonal hydroclimatic impacts of Sun Corridor expansion. *Environ Res Lett* 7(3):034026–034035.

Specific Comments

Lines 49...: “and heat mitigation strategies are assessed via their climatic impacts [8,9,10]. ”
In making this statement, the authors neglect important recent work that has addressed such heat mitigation strategies that extend to other sectors. For example, Georgescu et al. (2014) examined the efficacy of several such strategies by extending the heretofore climate discussion to the energy sector, and additionally compared such effects to emissions of LLGHGs. Such attribution of the existing literature is necessary to ensure a balanced perspective is offered.

Georgescu, M., Morefield, P. E., Bierwagen, B. G., & Weaver, C. P. (2014). Urban adaptation can roll back warming of emerging megapolitan regions. *Proceedings of the National Academy of Sciences*, 111(8), 2909-2914.

Additional research has demonstrated the synergistic interactions of urban-induced warming and heat waves, and given the Introductory message the authors try to convey, several recent

manuscripts come to mind as important to include as part of the discussion, including Li and Bou-Zeid (2013):

Li, D., & Bou-Zeid, E. (2013). Synergistic interactions between urban heat islands and heat waves: the impact in cities is larger than the sum of its parts*. Journal of Applied Meteorology and Climatology, 52(9), 2051-2064

We agree with the reviewer's comment that previous studies have investigated the cross-sector benefits of heat mitigation strategies. We did not intend to imply that the exiting literature on heat mitigation strategies ignores all cross-sectoral effects such as effects on energy, health, and greenhouse gas emissions. We have edited the manuscript to reflect this point and make it clear the link between water conservation and heat mitigation measures is the particular cross-sectoral gap that we are interested in addressing. We further included studies by Georgescu et al [2014] and Li et al [2013]:

L45: "While urban heat mitigation strategies have been shown to have beneficial effects on health, energy consumption, and greenhouse gas emissions [26,8,9,10,37], their implications for water conservation have not been widely examined."

L85: "Numerous studies, on the other hand, investigated the effects of heat mitigation strategies on health, energy consumption, and greenhouse gas emissions [26,8,9,10,37]. The effectiveness of cool roofs, in particular, has been broadly investigated as a promising heat mitigation measure [26] that shows potential to meaningfully decrease outdoor and indoor temperatures, reduce cooling loads, and offset CO₂ emission via negative global radiative forcing. There are compelling reasons to expect cross-sectoral impacts between water conservation and heat mitigation strategies. A few studies, for instance, have recently investigated the effect of water conservation measures on heat mitigation [27 and 29]. However, the implications of heat mitigation efforts for water conservation have not been widely examined. "

L65: "Warming trends and their potential consequences for energy demand and public health in urban areas are of high concern around the world [21-25, 38]."

[37] Georgescu, M., Morefield, P. E., Bierwagen, B. G., & Weaver, C. P. (2014). Urban adaptation can roll back warming of emerging megapolitan regions. *Proceedings of the National Academy of Sciences*, 111(8), 2909-2914.

[38] Li, D., & Bou-Zeid, E. (2013). Synergistic interactions between urban heat islands and heat waves: the impact in cities is larger than the sum of its parts*. *Journal of Applied Meteorology and Climatology*, 52(9), 2051-2064

Line 65: "and unsustainable groundwater extractions [18,19]."

The Sustainable Groundwater Management Act is aimed explicitly to address the unsustainable groundwater extractions the authors assume will continue into the future. No matter your perceptions about AB1739, etc., this is a clear example of past extraction does

not equal future extraction. The authors are encouraged to provide better-directed discussion that supports this important research.

The reviewer makes a valid point. At any rate, unsustainable groundwater extraction is more of a consequence of drought and water management failures rather than a driver. Our goal here was to highlight future drivers of water scarcity challenges, and so have removed the reference to unsustainable groundwater extraction.

Line 106: "... Fig. 1b and 1c ..."

Why do the authors present Figure 1b before Figure 1a; or Figure 2 before Figure 1a ... etc. The authors should pay attention to how they organized their figures such that it represents the actual flow of the manuscript.

We thank the reviewer for this comment and have re-organized the order of figures accordingly.

Figure Captions: Several figure captions have misspellings (e.g., "temperture" in Figure 5).

We thank the reviewer for this comment and have addressed the misspelled words.

REVIEWER #2 (REMARKS TO THE AUTHOR):

This paper studied an extremely important topic using well designed and thoroughly validated numerical experiments, which were built on the improvements made by the leading author into WRF-UCM. The idea is novel and is relevant to policy makers and the general public, and the results are interesting and understandable (even to the general public). The methodology is sound and the paper is well written. I like the paper very much and I think this is probably the first paper I ever reviewed for which I don't have any comments. Good job. I suggest publish as it is.

We thank the reviewer for this positive assessment of our manuscript. We too believe our results are novel and relevant to policy makers and the general public. However, to address the robustness concerns brought up by another referee, we extended our study domain to more regions, including all of the major metropolitan areas in California rather than only San Francisco Bay area. We further conducted our experiments over 15 years of 2001-2015, rather than only drought years of 2012-2014. Doing so we have obtained 15 years worth of remotely sensed data to improve the representation of green vegetation fraction and surface albedo over urban areas. We further acquired irrigation water consumption, reference evapotranspiration, and air temperature data for 15 years for a comprehensive validation of our improved model showing an accurate and reliable model performance. It is noteworthy that we did all these extra experiments while conserving the high spatial resolution of our modeling framework. We believe expanding the study domain and time in addition to our extensive efforts to represent the reality of the urban surfaces, vegetated fraction, and urban irrigation using real-time satellite information, irrigation water measurements, and high-

resolution land cover maps have made are results not only more robust and reliable but more interesting as we are now able to assess variability of water saving potential of cool roofs over 18 counties and 15 years.

To reflect the new experiments, we have made substantial changes to our manuscript. However, the conclusions of the original manuscript hold true and we were able to show the robustness of our results.

REVIEWER #3 (REMARKS TO THE AUTHOR):

This is a very interesting scientific work that merits to be published subject to the following improvements:

We thank the reviewer this positive assessment of our manuscript. We took every comment very seriously and used them to improve our study and paper. We address the comments individually below. We need to note that, to address the robustness concerns brought up by another referee, we extended our study domain to more regions, including all of the major metropolitan areas in California rather than only San Francisco Bay area. We further conducted our experiments over 15 years of 2001-2015, rather than only drought years of 2012-2014. Doing so we have obtained 15 years worth of remotely sensed data to improve the representation of green vegetation fraction and surface albedo over urban areas. We further acquired irrigation water consumption, reference evapotranspiration, and air temperature data for 15 years for a comprehensive validation of our improved model showing an accurate and reliable model performance. It is noteworthy that we did all these extra experiments while conserving the high spatial resolution of our modeling framework. We believe expanding the study domain and time in addition to our extensive efforts to represent the reality of the urban surfaces, vegetated fraction, and urban irrigation using real-time satellite information, irrigation water measurements, and high-resolution land cover maps have made are results not only more robust and reliable but more interesting as we are now able to assess variability of water saving potential of cool roofs over 18 counties and 15 years.

To reflect the new experiments, we have made substantial changes to our manuscript. However, the main conclusions of the original manuscript hold true and we were able to show the robustness of our results.

a) Ageing effects have to be discussed and considered. How aging affects the results?

We thank the reviewer for bringing up this important point. The albedo values for cool roofs that we used are achievable after three years of wear and tear, based on the EPA Energy Star roof product list: (http://downloads.energystar.gov/bi/qplist/roofs_prod_list.pdf?8ddd-02cf). To clarify the cool roof representation in our model Section 3 is added to the supplementary information.

L143:

“3. Cool Roof Representation in WRF-UCM

For the Cool-Roof scenario, the widespread deployment of cool roofs is represented by using increased roof albedos over all the buildings within our study domain, which replace the MODIS-based albedo values used in the Control scenario. We prescribe the roof albedos of commercial/industrial and residential buildings to 0.85 and 0.60, respectively, based on the EPA Energy Star roof product list: (http://downloads.energystar.gov/bi/qplist/roofs_prod_list.pdf?8ddd-02cf). The commercial/industrial cool roof albedo of 0.85 is selected to reflect the highest roof albedo achievable by coating (=0.88 for SOLARFLECT coating). Although our intention is to assess the maximum potential impacts associated with widespread deployment of cool roofs, we choose a less aggressive value of 0.6 for residential cool roof albedo to account for aesthetic preference of the residents. It is noteworthy that the cool roof albedo values used in this study are reported by the EPA Energy Star to be achievable after three years of wear and tear.”

b) An error and a sensitivity analysis have to be included.

We agree with the reviewer that statistical analysis and quantifying the uncertainties associated with the results are very important. To address this comment, we added error bars to Figure 2 and 4, representing the inter-annual variations of cool-roof effects over 2001-2015. Our results show that the cool-roof-induced effects on irrigation water demand and air temperature are persistent over the 15 years.

Figure 2. Simulated irrigation water consumption in gallons per capita per day (gpcd) for Control and Cool-Roof simulations for each county. Values represent averages over urban surfaces for June-October of 2001-2015. The error bars illustrate inter-annual fluctuations of irrigation water consumption reductions induced by cool roofs. County populations are represented in orange bars.

Figure 4. Simulated Cool-roof-induced changes in air temperature (a and b) and 2-m air temperature for Control and Cool-Roof simulations for each county (c). Values represent averages over June-October of 2001-2015. The error bars in (c) illustrate inter-annual fluctuations of air temperature changes induced by cool roofs. The solid black lines in (a and b) illustrate the boundaries of the 18 urban counties that are captured in the model domains. The boundaries of urban surfaces are illustrated by dotted black lines. Note that only changes that are statistically distinguishable from zero at 95% confidence interval are included.

Furthermore, as mentioned in the supplementary information (L76) and figure captions, we assess the statistical significance of the simulated changes, relative to model internal variability and natural variabilities of the climate system by applying the two-sided Student's t test to the model results. Analyzing the daily variations in each grid cell for the multiyear ensemble members, we only include the cool-roof-induced changes that are statistically significant with a 95% confidence level.

c) The results of the simulation have to be compared against similar simulations carried out for the same area, (Taha et al, Akbari, Menon, etc).

We thank the reviewer for this important comment. Following the reviewer's suggestion, we compared the cooling effects of cool roofs reported in our study with previous studies over Los Angeles areas in the manuscript (see below). However, as this is the first study to investigate

the water saving potential of a heat mitigation strategy we are not able to compare the water savings, induced by cool roofs, to any previous study.

L152: “The cooling effects of cool roofs reported here are in agreement with the previous studies in the Los Angeles area reporting cooling signals of 1-2.5 °C [10;39].”

[10] Vahmani, P., F., Sun, A., Hall, and G. Ban-Weiss (2016), Investigating the climate impacts of urbanization and the potential for cool roofs to counter future climate change in Los Angeles, *Environ. Res. Lett.*, accepted (Article reference: ERL-102897).

[39] Akbari H. *Saving energy and improving air quality in urban heat islands*. AIP Conference Proceedings 1044, 192 (2008); doi: <http://dx.doi.org/10.1063/1.2993720>

d) The basic physics mechanism on the relation between high albedo and water conservation has to be better explained.

To address this concern of the reviewer we revised the manuscript to include the followings:

L173: “The increased surface albedo, induced by cool roofs, results in increased reflected solar radiation and therefore decreased surface energy budget, decreased turbulent fluxes, and finally lower ET or evaporative water demand rates.”

L129: “Evapotranspiration (ET), or evaporative water demand, is the main driver of urban irrigation consumption, explaining 91 and 92% of irrigation water variations across urban areas in NorCal and SoCal, respectively (Supplementary Fig. S9). Therefore, to better understand irrigation water demand and its response to cool-roofs, we next focus on identifying the drivers of spatial and temporal variabilities of ET.”

L144: “The strong relationship between daily evaporative water demand and air temperature, reported above, illustrates that cooling effects of heat mitigation measures, such as cool roofs, can in turn lead to reduced ET and evaporative water demand and therefore reduced irrigation water use.”

REVIEWERS' COMMENTS:

Reviewer #1 (Remarks to the Author):

Review for – Nature Communications Manuscript Number: NCOMMS-17-00243A
Title: Water conservation benefits of urban heat mitigation

I applaud the authors for significantly improving their work by (i) increasing sampling of summers that are investigated and (ii) including extended portions of California whereby the complexity of varied geographies is accounted for. These were the two principal reservations I had, and the authors have addressed them clearly and eloquently.

I have but a few simple comments, and after the necessary revisions are made, I very much recommend publication of the manuscript in Nature Communications.

Specific Comments

1. Line 51:

The authors state they have conducted multi-year simulations, but instead, they have conducted 5-month simulations, repeated 15 times. Please rephrase appropriately (e.g., 15 summers) as it leads someone to believe continuous simulations have been performed.

2. Line 98: "... we employ a customized and validated Weather..."

Please change to a customized and validated version of the Weather ..."

3. Line 110: "... to of ..."

Delete "of" after "to".

4. Lines 117 ~ 120:

As was done for SF county and LA county in the text below, the Marin and Riverside urban irrigation water use per capita should be stated, for consistency, but also to illustrate the disparity between these counties and the ones mentioned below.

5. Lines 141-142: "... where wind generates turbulence and transports hotter air into urban regions

..."

The snippet above is confusing and should be rephrased, with an improved process-based discussion (1-2 sentences) that explains wind/ET correlation. On the scales investigated here, it is turbulence that is responsible for energy and mass transport, not the other way around as the authors state (i.e., over large scales and above the PBL, Coriolis and PGF drive air flow, but within the PBL, turbulence plays the dominant role rather than the mean wind driving everything).

6. Line 169: "... religions ..."

Probably meant to state "regions".

7. Line 195-196: "... medium density regions benefit more in terms of reduced evaporative water demand."

The assumption here is that the entirety of these permeable surfaces are irrigated, which is unlikely. Best to introduce this caveat in the text.

8. Line 241: "...in regions with lower cloud cover ...".

Low cloud cover could be interpreted as having a low ceiling base. Probably the authors should state ' "reduced" cloud cover ' rather than "lower cloud cover", to avoid misunderstanding.

9. Line 399:

Although the authors state so in the supplementary document, they should probably also indicate

the version of the WRF model they are using on line 399, when first describing the model.

10. Figure Captions: Unfortunately, several figure captions (in the main paper and supplement) continue to have misspellings throughout (e.g., "boundries" in Figure 1).

Reviewer #3 (Remarks to the Author):

The authors ahve replied adequantly my comments

Reviewer #1 (Remarks to the Author):

Review for – Nature Communications Manuscript Number: NCOMMS-17-00243A

Title: Water conservation benefits of urban heat mitigation

I applaud the authors for significantly improving their work by (i) increasing sampling of summers that are investigated and (ii) including extended portions of California whereby the complexity of varied geographies is accounted for. These were the two principal reservations I had, and the authors have addressed them clearly and eloquently.

I have but a few simple comments, and after the necessary revisions are made, I very much recommend publication of the manuscript in Nature Communications.

We thank the reviewer for the constructive suggestions and comments. We too believe addressing these comments and expanding our study domain and time period has improved the robustness of our results.

Specific Comments

1. Line 51:

The authors state they have conducted multi-year simulations, but instead, they have conducted 5-month simulations, repeated 15 times. Please rephrase appropriately (e.g., 15 summers) as it leads someone to believe continuous simulations have been performed.

We thank the reviewer for this comment. To address this issue, we removed ‘multiyear’ from the abstract as there was not enough room for us to explain what we mean by multiyear there.

We clearly state the time frame of the simulations later in the manuscript:

“We simulate the warm, dry months of June-October for 15 years (2001-2015) under two scenarios...”

2. Line 98: “... we employ a customized and validated Weather...”

Please change to a customized and validated version of the Weather ...”

We agree with the proposed change and made the change in the revised manuscript.

3. Line 110: “ ... to of ...”

Delete “of” after “to”.

We agree with the proposed change and made the change in the revised manuscript.

4. Lines 117 ~ 120:

As was done for SF county and LA county in the text below, the Marin and Riverside urban irrigation water use per capita should be stated, for consistency, but also to illustrate the disparity between these counties and the ones mentioned below.

We thank the reviewer for this suggestion. The text is edited to include the irrigation water use per capita for Marin and Riverside counties as well:

“We further show that Marin county in NorCal and Riverside county in SoCal have highest irrigation water demand per capita of 373 and 257 gallons per day, respectively, due to a high fraction of low intensity residential areas and associated vegetation cover (urban fractions of 0.24 and 0.26, respectively).”

5. Lines 141-142: “... where wind generates turbulence and transports hotter air into urban regions ...”

The snippet above is confusing and should be rephrased, with an improved process-based discussion (1-2 sentences) that explains wind/ET correlation. On the scales investigated here, it is turbulence that is responsible for energy and mass transport, not the other way around as the authors state (i.e., over large scales and above the PBL, Coriolis and PGF drive air flow, but within the PBL, turbulence plays the dominant role rather than the mean wind driving everything).

We agree with the reviewer and edited the manuscript to explain the link between wind speed and ET over inland counties:

“This analysis also shows that increased wind speed results in increased ET. This relationship is the strongest over inland counties (e.g., Napa, San Joaquin, and Stanislaus), where wind transports hot and dry air into urban regions (Supplementary Fig. S11), providing energy for evapotranspiration and, at the same time, increasing evaporative water demand by reducing vapor pressure (i.e., partial pressure of water vapor) in the atmosphere.”

6. Line 169: “... religions ...”

Probably meant to state “regions”.

We are sorry about this mistake and replaced the misspelled word.

7. Line 195-196: “... medium density regions benefit more in terms of reduced evaporative water demand.”

The assumption here is that the entirety of these permeable surfaces are irrigated, which is unlikely. Best to introduce this caveat in the text.

We agree with the reviewer’s comment and added the following text to the mentioned discussion to point out the assumption that all the pervious surfaces in urban areas are assumed to be irrigated:

“It is noteworthy that we assume the entirety of pervious surfaces in urban areas is irrigated. This assumption might affect the role of urban fraction, as discussed above, in determining baseline and reduced ET rates.”

8. Line 241: “...in regions with lower cloud cover ...”.

Low cloud cover could be interpreted as having a low ceiling base. Probably the authors should state ‘ “reduced” cloud cover ’ rather than “lower cloud cover”, to avoid misunderstanding.

We thank the reviewer for this suggestion. The text is edited accordingly.

9. Line 399:

Although the authors state so in the supplementary document, they should probably also indicate the version of the WRF model they are using on line 399, when first describing the model.

We thank the reviewer for this comment. The text is edited accordingly.

10. Figure Captions: Unfortunately, several figure captions (in the main paper and supplement) continue to have misspellings throughout (e.g., “boundries” in Figure 1).

We are sorry about this mistake and replaced the misspelled word.